# Optimization and Renovation Design of Indoor Thermal Environment in Traditional Houses in Northeast Sichuan (China)—A Case Study of a Three-Section Courtyard House

Chaoping Hou [1,2,*], Wentao Hu [3], Yuefan Jiang [2] and Weijun Gao [1,*]

1  Faculty of Environmental Engineering, The University of Kitakyushu, Kitakyushu 808-0135, Japan
2  College of Architecture and Urban-Planning, Sichuan Agricultural University, Chengdu 611830, China; jiangyuefang@yeah.net
3  Institute of Civil Engineering and Architecture, Ural Federal University, 19, Mira St., Yekaterinburg 620002, Russia; huwentaogood@163.com
*  Correspondence: hchaoping@sicau.edu.cn (C.H.); gaoweijun@me.com (W.G.)

**Abstract:** The three-section courtyard is the most representative traditional residence in the traditional villages in northeast Sichuan. As a unique cultural landscape, it carries the local historical style and cultural connotation. However, the high temperature weather in summer leads to a poor thermal environment in traditional residential buildings, which cannot meet the needs of building users for human thermal comfort, and the wall is the most critical factor affecting the indoor thermal environment. Therefore, to optimise the indoor thermal environment of traditional residential buildings, this study designed four groups of wall renovation schemes according to the original traditional residential buildings and modern technology, and simulated and verified the feasibility of the building renovation schemes by using Design Builder. Then, the four groups of wall renovation schemes were compared and tested based on the Design Builder. Comparative results of the thermal-performance evaluation index revealed that compared with Case 1 and Case 2, the building refrigeration energy consumption of Case 3 in the hottest week was the least, only 427.7 kW·h, which indicates that the external wall renovation scheme using aerated concrete blocks had the best thermal insulation and energy-saving effects. The cooling energy consumption of Case 4 in the hottest week was 422 kW·h, which was 4.3 kW·h less than that of Case 3, indicating that the wall renovation scheme with an air inter-layer had better thermal insulation and energy-saving effects. The refrigeration energy consumption of Case 7 in the hottest week was only 409.8 kW·h, which was 4.19% lower than Case 3 (without insulation material), indicating that the scheme of selecting central insulation and extruded polystyrene board (XPS) had better thermal insulation and energy-saving effects in practical projects. In summary, the above transformation scheme not only improves the indoor thermal environment of traditional residential buildings, but also provides guidance for architectural designers on green, energy-saving and sustainable design.

**Keywords:** traditional residence; three-section courtyard; indoor thermal environment; software simulation; field research; comparative analysis method

## 1. Introduction

As a major part of the cultural landscape, traditional houses have local historical features and cultural connotations [1]. Traditional dwellings are often complementary to the cultural environment of the region; therefore, the protection and study of traditional dwellings is also the protection and study of the entire traditional village to a certain extent [2,3]. As a carrier of history and culture, architecture is constantly updated throughout history. It is not only a product of a single period but also a collection of multiple historical periods and human environments [3]. Therefore, studying, inheriting, and developing traditional houses are extremely important [4–6].

With the increasing demand from residents for thermal comfort, balancing energy consumption in built environments is a great challenge. Climate adaptation strategies have been widely applied to the design of traditional dwellings worldwide, and climate adaptation measures adopted by traditional dwellings have been extensively developed [7,8]. Therefore, the advantages of traditional residential buildings should be retained, and optimisation strategies should be proposed to make use of climate and weather conditions and to make reasonable use of solar radiation [9], natural ventilation [10], and other renewable resources to improve the indoor thermal environment and save building energy [11].

Climate adaptation strategies have become the primary measure for traditional building renovations worldwide [12]. The climate adaptation measures adopted by traditional buildings include choosing reasonable building forms and site locations, optimising building structures, improving indoor natural ventilation, and renovating suitable roofs. Currently, many researchers are engaged in related research [13–15]. Owing to the advancement of science and technology, problems existing in traditional buildings are constantly amplified, and the active improvement of indoor thermal environments in traditional buildings is becoming increasingly common. Unfortunately, common methods of improving the quality of life in environments with high energy consumption eliminate the advantages of traditional buildings [16]. However, with the aggravation of the energy crisis and greenhouse effect, researchers have begun to study climate adaptation strategies for traditional buildings and improve the quality of the living environment through smart design and the rational use of natural resources [17].

Several researchers have optimised the indoor thermal environments of traditional buildings. J Hou et al. [18]. added insulation materials with optimal thickness to the exterior walls of traditional rural dwellings in the mountainous areas of northeast Sichuan to improve their thermal performance, improve the indoor thermal environment and reduce energy consumption. Using degree-day method and P1-P2 economic model, the optimum thicknesses of five kinds of insulation materials were calculated, and the energy-saving and economic benefits were evaluated based on EnergyPlus and the dynamic payback period model. The results show that under the local climate and economic conditions, the optimal insulation thickness is 0.081~0.144 m. Reasonable reduction of heating days can increase energy-saving rate and economic benefit by 29.5% and 78.2%, respectively. In summary, adding insulation materials with the best thickness to the external walls of traditional dwellings is conducive to improving their thermal performance, energy-saving rate and economic benefits. However, there is a lack of comprehensive consideration of factors such as filling location and insulation material type.

L Gou et al. [19]. built two identical model buildings, one of which used 38 mm expanded polystyrene (EPS) panels as exterior insulation. Under the same heating set point temperature, the dynamic change process of the indoor thermal environment and heating energy consumption of an EPS-insulated building and non-insulated reference building were compared and monitored. The results showed that under the same heating set point temperature, the indoor temperature of the two buildings can quickly approach the set point value in about 20 min, but the temperature rise of the inner surface of the wall lags behind. The energy consumption of the two types of buildings in the initial 20 min heating period was similar, and the energy-saving rate of EPS buildings increased slowly with the increase in heating time. Intermittent heating for 0.5 to 8 h in hot summer and cold winter areas has a power saving rate of about 1% to 26%. In summary, such expanded polystyrene insulation material has a good effect on the indoor thermal environment, but there is a lack of research on the effect of filling position and wall main material on indoor thermal environment.

Z Li et al. [20]. broke through the original thermal design idea, which mainly aimed at improving the thermal insulation performance of building walls, and proposed a thermal design method for building walls that considered both local climate conditions and thermal insulation. Based on the typical weather data in winter, the thermal insulation performance of building walls was calculated for rooms without heating measures in Lhasa. The results

show that there is great potential to improve the indoor thermal environment in Lhasa by combining solar energy-rich climate conditions with thermal insulation technology. Considering the "carrier" effect of the wall on the available climate resources, a higher the insulation performance, does not necessarily lead to a better effect. In summary, the results show that the technology with a better thermal insulation effect may not be adaptable to the climatic conditions of different places. Therefore, the climatic conditions of northeast Sichuan region must be considered to improve the thermal insulation performance and indoor thermal environment.

Liting Yuan et al. [21] studied the optimal thermal insulation position of external walls to reduce building energy consumption. This paper analyses the transmission load generation of external insulation and internal insulation and external walls, and evaluates the parameters that may affect the energy-saving advantages of the two insulation locations. An analytical model is used to calculate the heat transfer of the wall. The results show that the heat storage of the external wall during non-use is the key to determining the relative energy saving of the insulation position under the condition of an intermittent convection air system. In summary, exploring the best thermal insulation position and improving the heat storage capacity of the external wall are important factors for improving the building energy efficiency and indoor thermal environment. Therefore, there is an urgent need for a design model that improves the search for an optimal thermal insulation location and improves the thermal storage capacity of external walls.

In addition to an analysis and exploration of previous literature, Hou et al. [22] also investigated the climate characteristics of northeast Sichuan and the indoor thermal environment of in situ traditional dwellings. As shown in Figure 1, the outdoor temperature peaked at 34.2 °C at 14:00, and 30.3 °C was the recorded temperature for the living room and two bedrooms at 15:00. The peak temperature of each room decreased and was delayed relative to the outdoor temperature. The temperature increased gradually during the day, whereas the temperature decreased gradually at night. The change trend of the indoor temperature was delayed compared to that of the outdoor temperature. This phenomenon shows that the temperature of traditional residential buildings is higher in summer, and the average indoor air temperatures of the living room and two bedrooms exceed the comfortable temperature range of 22.8–28.8 °C. Only natural ventilation at night can contribute to temperature reduction. Therefore, the thermal insulation effect of traditional residential buildings in summer is poor and cannot meet the thermal comfort needs of building users.

In summary, previous studies show that researchers from various countries have transformed traditional residential buildings through modern technologies, such as new materials, insulation technology for envelope structures, and cooling technology, and have achieved positive effects in improving the indoor thermal environment. However, a field investigation in northeast Sichuan showed that the high temperature in summer leads to a poor indoor thermal environment in traditional residential buildings, which cannot meet the needs of building users for human thermal comfort. Therefore, to optimise the indoor thermal environment of traditional residential buildings, this study designed four groups of wall renovation schemes according to the original traditional residential building design and modern technology, and simulated and verified the feasibility of the schemes using DesignBuilder (Version 7). This is expected to improve the indoor thermal environment of traditional residential buildings, and get a building renovation scheme that meets the comprehensive needs of traditional residential buildings in northeast Sichuan, so as to provide guidance for architectural designers on green, energy-saving.

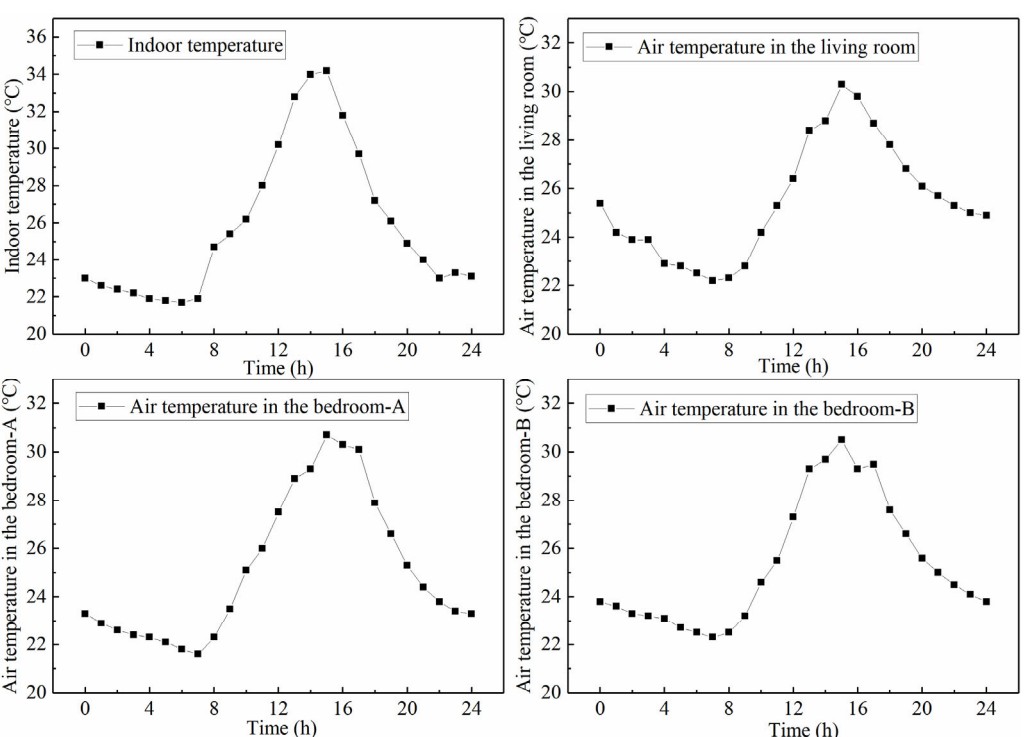

**Figure 1.** Diagram of indoor and ambient temperature curves.

## 2. Methodology

### 2.1. General Situation and Climatic Conditions of Traditional Houses in Northeast Sichuan

The object of this study was Liyuanba Village, which is the most representative and best-preserved traditional village. Figure 2 shows the location analysis diagram of northeastern Sichuan and Liyuanba Village. Northeastern Sichuan is hot in the summer and cold in the winter. The northeastern area of Sichuan is surrounded by mountains and hills, with a significant drop in altitude. The climate is characteristically wet and rainy, with a low solar rate and annual precipitation of more than 1000 mm, mainly concentrated in the summer from June to September. Summer is hot and humid, with temperatures above 30 °C in the hottest months and an average relative humidity of approximately 77%. Winters are wet and cold, with an average temperature of 4.1 °C in the coldest months and an average relative humidity of approximately 68%. In summary, the local residents require cooling in summer and heating in winter.

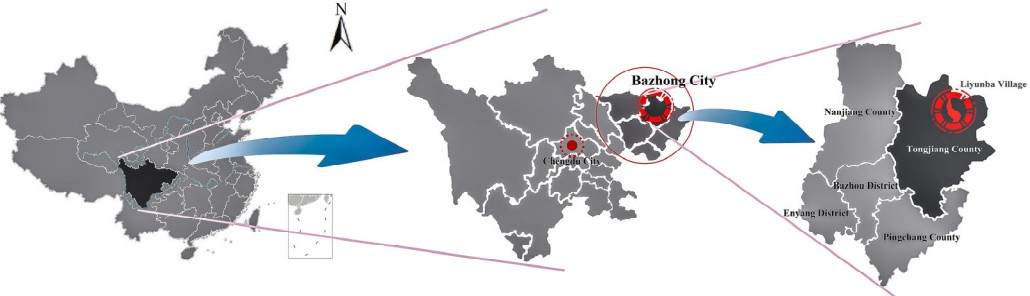

**Figure 2.** Location analysis diagram of northeast Sichuan and Liyuanba Village.

Figure 3 illustrates the general situation of traditional dwellings in northeast Sichuan. Most traditional villages in northeast Sichuan are located on sunny slopes in the middle of the mountains. The mountains and water bodies must also be analysed. Adequate water resources are essential for agricultural production. Villagers' fields lie at the foot of the mountains on both sides of the water system, which solves the problem of agricultural irrigation. Most of the villages are near mountains and water and are integrated with nature,

which helps improve the comfort of the building users. Most houses in northeastern Sichuan are perforated wooden structures with green tile roofs. For the lighting requirements, sliding or fixed windows are designed and installed on the walls. The sandwiching walls are made of multipurpose bamboo braids, external grass mud, and lime paint.

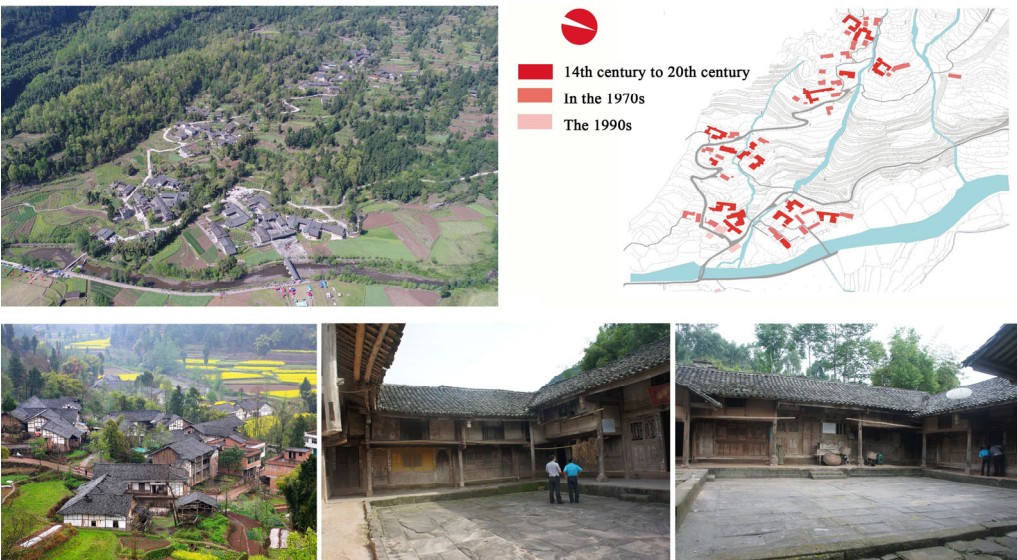

**Figure 3.** General situation of traditional houses in northeastern Sichuan.

## 2.2. Mathematical Model and Thermal Physical Parameters of Traditional Residential Buildings

### 2.2.1. Establishment and Parameter Settings of Traditional Residence Model

DesignBuilder is a comprehensive user-friendly graphical interface simulation software developed for EnergyPlus (a building energy dynamic simulation program) for building heating, cooling, lighting, ventilation, daylighting, and other popular total energy simulations and economic analyses. In this study, DesignBuilder was used to simulate the indoor temperature and building energy consumption of traditional residential buildings in northeast Sichuan. The influence of each parameter on the indoor thermal environment was analysed simultaneously, and the effectiveness of various improvement strategies for traditional residential buildings in northeast Sichuan was verified. Two factors affecting the energy consumption of residential refrigeration systems are external and internal disturbances. External disturbances primarily refer to the influence of outdoor meteorological conditions. Internal disturbances refer to the influence of indoor personnel activities and the equipment in a room. However, because the purpose of this study was to analyse the impact of building structures on building energy consumption, the software setting ignored the impact of internal personnel and equipment on building energy consumption.

As shown in Figure 4(1), the traditional residence (courtyard house) under investigation was taken as the simulation object, and a model of the traditional residence was established using simulation software. Figure 4(2) shows the plan of the traditional residence and the location of the experimental room. The red area is the entire test area; the living room in the middle is Room A; and the bedrooms on both sides are Rooms B and C. The living room and bedrooms of traditional folk houses were selected as the representative experimental rooms in this experiment, since they are the spaces that residents use most frequently and for the longest time in daily life. In addition, in traditional residential houses, refrigeration spaces are located in the living room and bedrooms. A typical summer high-temperature climate (15 June to 31 August) was selected to simulate energy consumption and refrigeration temperature.

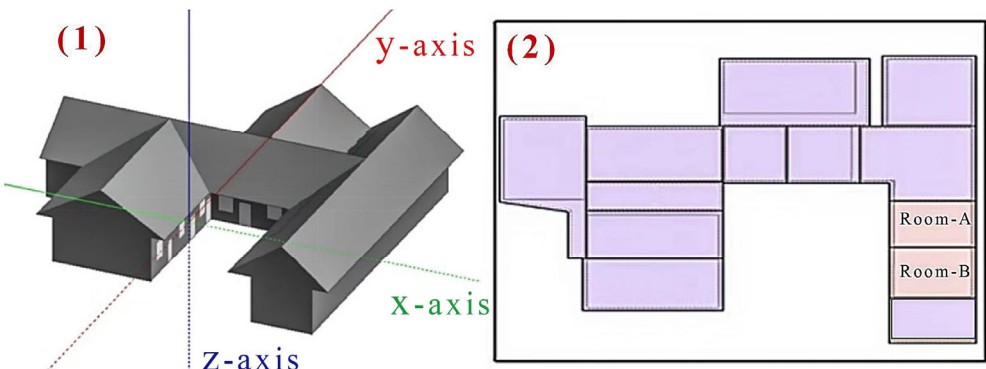

**Figure 4.** (**1**) The model of the traditional residential house and (**2**) the floor plan of the experimental room.

2.2.2. Heat Transfer Analysis of Traditional Residential Building Model

In this experiment, the energy consumption simulation software performed the joint calculation of multiple physical and mathematical models. In the software, the heat transfer process of the traditional residential house model was divided into three parts: the physical and mathematical model of the outdoor surface heat balance, the heat transfer mathematical model of the wall, and the physical and mathematical model of the indoor heat balance.

(1) There is a functional relationship between the heat transfer process of the outdoor surface and solar radiation, the surrounding environment, and outdoor air. The expressions for the physical and mathematical models of the heat balance of the outdoor surface are as follows:

$$q''_{\alpha sol} + q''_{LWR} + q''_{conv} + q''_{ko} = 0 \qquad (1)$$

(2) The heat transfer process for the exterior wall of the building was divided into three stages:

  (a) When the air temperature of the room ($T_{in}$) is higher than that of the wall inner surface ($T_j$), that is, $T_{in} > T_j$, Equation (2), the heat absorbed by the inner surface of the wall ($q_i$), is given by

$$q_i = \alpha_i(T_{in} - T_j) \qquad (2)$$

  (b) When the temperature of the inner surface of the wall ($T_j$) is higher than that of the outer surface ($T_i$), that is, $T_j > T_i$, the heat conduction (Equation (3)) of the wall material layer ($q_y$) is

$$q_y = \frac{T_j - T_i}{R} \qquad (3)$$

  (c) When the temperature of the outer wall surface ($T_i$) is higher than the outer air temperature ($T_{out}$), that is, $T_i > T_{out}$, Equation (4), which is the heat released by the wall's outer surface, ($q_e$) is

$$q_e = \alpha_e(T_i - T_{out}) \qquad (4)$$

(3) The physical and mathematical models of the indoor heat balance are generally modelled using four coupled heat transfer components: conduction through building units, air convection, short-wave radiation absorption and reflection, and long-wave radiation exchange. The incoming short-wave radiation originates from solar radiation entering the area through the window and the emittance of the internal light source. Long-wave radiation exchange includes the absorption and emission of low-temperature radiation sources such as surfaces, devices, and people in all other areas. In summary, the physical and mathematical models of the indoor heat balance can be expressed as

$$q''_{LWX} + q''_{SW} + q''_{LWS} + q''_{ki} + q''_{ki} + q''_{sol} + q''_{conv} = 0 \tag{5}$$

### 2.2.3. Physical Parameters of Building Materials

The simulation data were obtained by simulating the measured outdoor climate data. In this study, DesignBuilder was used to conduct the simulation research. The building materials used in the research model were the building materials of actual buildings, and the shape and size of typical traditional houses were simulated. The results obtained by simulating the traditional residence model further verify the accuracy of the measurement data. In addition, a model simulation was performed for the subsequent optimisation strategy. The physical parameters of the envelope structure of traditional residential buildings are listed in Table 1.

**Table 1.** Physical parameters of the building structure.

| Name | Materials | Thickness (mm) | Density (kg/m³) | Specific Heat [J/(kg·k)] | Heat Conductivity Coefficient [W/(m·k)] |
|---|---|---|---|---|---|
| Roof | Tile | 10 | 1920 | 1260 | 1.59 |
| | Clay | 35 | 1600 | 1010 | 0.76 |
| | Board | 15 | 500 | 2510 | 0.17 |
| Bamboo and earth walls | Calcicoater | 10 | 1800 | 1050 | 0.81 |
| | Rammed earth | 60 | 1795.6 | 884 | 0.72 |
| | Calcicoater | 10 | 1800 | 1050 | 0.81 |
| Wooden exterior wall | Board | 50 | 500 | 2510 | 0.17 |
| Door | Board | 50 | 500 | 2510 | 0.17 |
| Floor | Limestone soil | 100 | 2000 | 1010 | 1.16 |
| | Stone block | 300 | 2400 | 920 | 2.04 |
| Ceiling | Board | 50 | 500 | 2510 | 0.17 |
| Interior wall | Calcicoater | 10 | 1800 | 1050 | 0.81 |
| | Rammed earth | 10 | 1795.6 | 884 | 0.72 |
| | Calcicoater | 10 | 1800 | 1050 | 0.81 |

### 2.3. Renovation Plan of Traditional Houses

Traditional houses in northeast Sichuan are generally wooden structures; their external walls are usually bamboo earth walls, although a few are brick walls. In this experiment, the walls of the traditional dwellings were typical bamboo–earth sandwich walls. The construction of bamboo–earth sandwich walls is as follows: First, the wooden structure of a traditional dwelling is filled with bamboo–earth sandwich walls. A soil coating is then applied to both sides of the sandwich walls to level the surfaces, and a lime coating is applied after the soil coating dries. This helps the walls combine well with the local climate characteristics, resulting in a high porosity, light weight, low heat capacity, and resistance to moisture and corrosion.

Based on the main wall material, air-layer insulation, insulation materials, and different insulation positions, this experiment optimised the wall design of traditional residential buildings and evaluated high-quality transformation schemes to improve the insulation performance of walls and indoor thermal environments. In this experiment, eight transformation schemes were designed, which were divided into four experimental groups for simulation, comparison, and analysis.

(1) Renovation schemes of different primary wall materials

The thermal performance of the primary wall material has a significant impact on the indoor thermal environment. Therefore, three renovation schemes for the primary wall material were designed based on solid clay brick, hollow brick, and aerated concrete. Table 2 lists the optimal design schemes for the primary materials used in traditional residential walls.

**Table 2.** Reconstruction scheme of the primary wall materials.

| Case | Materials (Arranged from Inside to Outside) | Thickness (mm) | Density (kg/m$^3$) | Specific Heat [J/(kg·k)] | Heat Conductivity Coefficient [W/(m·k)] |
|---|---|---|---|---|---|
| Case 1 | Cement mortar | 10 | 0.93 | 1050 | 1800 |
| | Solid clay brick | 200 | 0.76 | 1086 | 1700 |
| | Cement mortar | 10 | 0.93 | 1050 | 1800 |
| Case 2 | Cement mortar | 10 | 0.93 | 1050 | 1800 |
| | Porous clay brick | 200 | 0.58 | 1062 | 1400 |
| | Cement mortar | 10 | 0.93 | 1050 | 1800 |
| Case 3 | Cement mortar | 10 | 0.93 | 1050 | 1800 |
| | Aerated concrete block | 200 | 0.18 | 1430 | 400 |
| | Cement mortar | 10 | 0.93 | 1050 | 1800 |

(2) Wall reconstruction scheme of added air interlayer

Closed-air interlayer insulation walls are widely used in rural buildings, as they not only save wall building materials but also have a very small heat transfer coefficient of air because of the closed-air interlayer inside and good heat insulation performance. Table 3 lists the wall renovation schemes with an added air interlayer.

**Table 3.** Modification scheme of adding air interlayer.

| Case | Materials (Arranged from Inside to Outside) | Thickness (mm) | Density (kg/m$^3$) | Specific Heat [J/(kg·k)] | Heat Conductivity Coefficient [W/(m·k)] |
|---|---|---|---|---|---|
| Case 3 | Cement mortar | 10 | 0.93 | 1050 | 1800 |
| | Aerated concrete block | 200 | 0.18 | 1430 | 400 |
| | Cement mortar | 10 | 0.93 | 1050 | 1800 |
| Case 4 | Cement mortar | 10 | 0.93 | 1050 | 1800 |
| | Aerated concrete block | 100 | 0.18 | 1430 | 400 |
| | Air space | 20 | 0.026 | 1005 | 1210 |
| | Aerated concrete block | 100 | 0.18 | 1430 | 400 |
| | Cement mortar | 10 | 0.93 | 1050 | 1800 |

(3) Adding insulation materials to the wall transformation scheme

The thermal insulation structure in the wall not only provides good heat insulation but also protects the wall and extends its service life. External wall insulation technology is preferred in the northeastern Sichuan Province. Therefore, this experiment adopted an expanded polystyrene board (EPS), an extruded polystyrene board (XPS), and a mineral wool board to transform traditional residential buildings. Table 4 shows the wall renovation scheme with added insulation materials.

**Table 4.** Modification scheme of thermal insulation materials.

| Case | Materials (Arranged from Inside to Outside) | Thickness (mm) | Density (kg/m³) | Specific Heat [J/(kg·k)] | Heat Conductivity Coefficient [W/(m·k)] |
|---|---|---|---|---|---|
| Case 5 | Cement mortar | 10 | 0.93 | 1050 | 1800 |
| | Aerated concrete block | 100 | 0.18 | 1430 | 400 |
| | Mineral wool board | 20 | 0.045 | 1220 | 140 |
| | Aerated concrete block | 100 | 0.18 | 1430 | 400 |
| | Cement mortar | 10 | 0.93 | 1050 | 1800 |
| Case 6 | Cement mortar | 10 | 0.93 | 1050 | 1800 |
| | Aerated concrete block | 100 | 0.18 | 1430 | 400 |
| | EPS | 20 | 0.042 | 1380 | 18 |
| | Aerated concrete block | 100 | 0.18 | 1430 | 400 |
| | Cement mortar | 10 | 0.93 | 1050 | 1800 |
| Case 7 | Cement mortar | 10 | 0.93 | 1050 | 1800 |
| | Aerated concrete block | 100 | 0.18 | 1430 | 400 |
| | XPS | 20 | 0.03 | 1380 | 20 |
| | Aerated concrete block | 100 | 0.18 | 1430 | 400 |
| | Cement mortar | 10 | 0.93 | 1050 | 1800 |

(4)  Wall renovation schemes for different thermal insulation positions

Because the filling position of the thermal insulation material significantly affects the thermal insulation performance of the wall, we designed a renovation scheme to evaluate the placement of thermal insulation in the middle compared to the inside of the wall. Table 5 lists the wall renovation scheme for the middle and inner insulations.

**Table 5.** Renovation plans for external and internal heat preservation.

| Case | Materials (Arranged from Inside to Outside) | Thickness (mm) | Density (kg/m³) | Specific Heat [J/(kg·k)] | Heat Conductivity Coefficient [W/(m·k)] |
|---|---|---|---|---|---|
| Case 7 | Cement mortar | 10 | 0.93 | 1050 | 1800 |
| | Aerated concrete block | 100 | 0.18 | 1430 | 400 |
| | XPS | 20 | 0.03 | 1380 | 20 |
| | Aerated concrete block | 100 | 0.18 | 1430 | 400 |
| | Cement mortar | 10 | 0.93 | 1050 | 1800 |
| Case 8 | Cement mortar | 10 | 0.93 | 1050 | 1800 |
| | XPS | 20 | 0.03 | 1380 | 20 |
| | Aerated concrete block | 200 | 0.18 | 1430 | 400 |
| | Cement mortar | 10 | 0.93 | 1050 | 1800 |

*2.4. Measuring Platform and Measuring Instruments*

In this experiment, the living rooms and bedrooms of traditional residential houses (three-section courtyards) were selected as experimental rooms, and field experimental data were collected from 15 June to 15 August. Figure 5 shows images of the field experiment. Based on field measurements, parameters such as the outdoor air temperature, relative humidity, and solar radiation intensity in northeastern Sichuan were obtained.

Figure 6 shows the indoor and outdoor measurement points of the traditional dwellings. The measurement points of the black-globe temperature, air temperature, and relative humidity were placed in the bedrooms and living room. The monitoring point was located 1.5 m from the ground vertically, and the horizontal plane was set in the centre of the plane. A monitoring point for air temperature and relative humidity was set up in the courtyard, located 1.5 m above the ground.

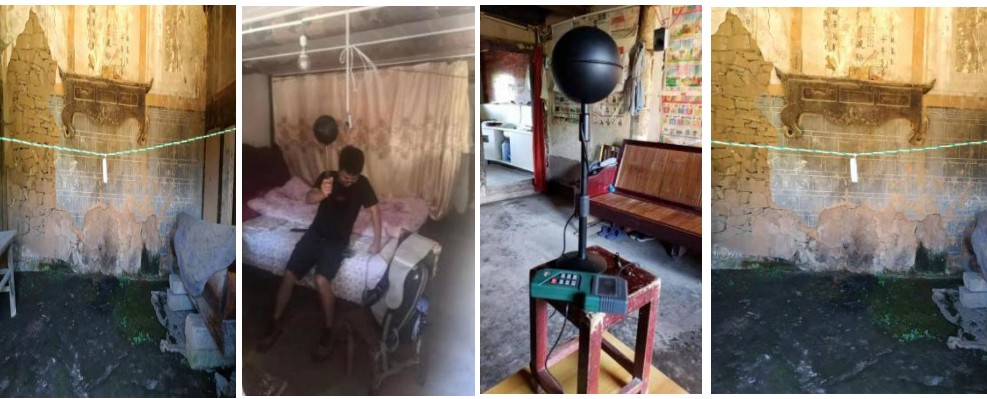

**Figure 5.** Traditional folk house field experiment.

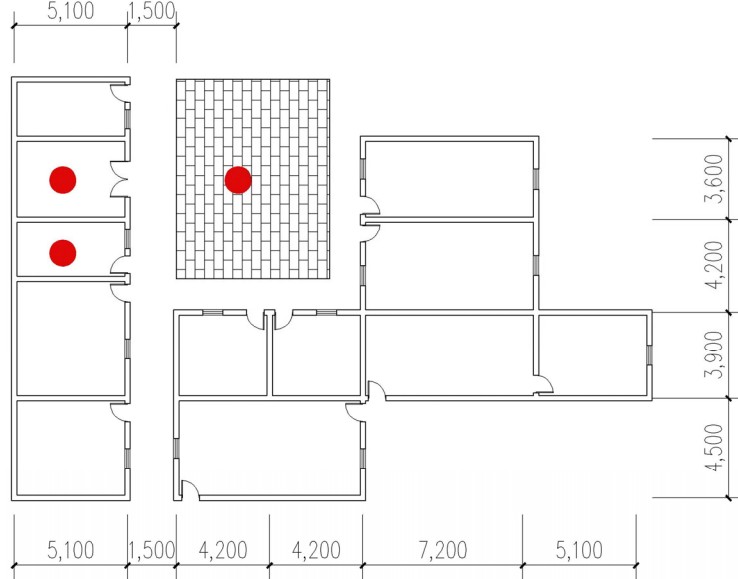

**Figure 6.** Layout of the locations of the measurement instruments in the traditional house.

By setting the sampling times of the automatic hygrograph and black globe thermometer, the air temperature and relative humidity were automatically recorded every 30 min, and the black globe thermometer reading was automatically recorded every 30 min. During the measurements, the bedroom wind speed was manually recorded every 30 min. Table 6 lists the technical parameters of the instruments, including the measurement range, accuracy, resolution, and response time.

**Table 6.** Experimental instruments and their parameters.

| Equipment Name | Type Specification | Measurable Variable | Measuring Range and Accuracy |
|---|---|---|---|
| Temperature and humidity tester | ZA88289 | Humidity | 0–100% (±3%) |
| | | Temperature | −40–85 °C (± 0.5 °C) |
| Solar radiation tester | JTR05 | Radiation | 0–2000 W/m$^2$ (±1 W/m$^2$) |
| | | Temperature | −40 °C–120 °C (±0.5 °C) |
| Wind speed tester | JT1402 | Wind speed | 0–20 m/s (±0.03 m/s) |
| Black globe temperature tester | JTR04 | Black globe temperature | −20–125 °C (±0.5 °C) |

## 3. Results and Discussion

### 3.1. Comparison and Verification between Measured Data and Simulated Data

The measured climate data for northeast Sichuan in summer were input into Design-Builder to simulate the indoor air temperatures of the bedroom and living room. Figure 7a presents the measured and simulated data for the bedrooms of traditional residential houses. The measured and simulated numerical curves for the bedroom were generally similar, with the highest indoor air temperature appearing at 15:00 and the lowest at 7:00. The simulated value of the average air temperature in the bedroom is 26.4 °C, and the measured value of the average air temperature in the bedroom is 25.8 °C. The relative error between the simulated and measured values is only 2.3%.

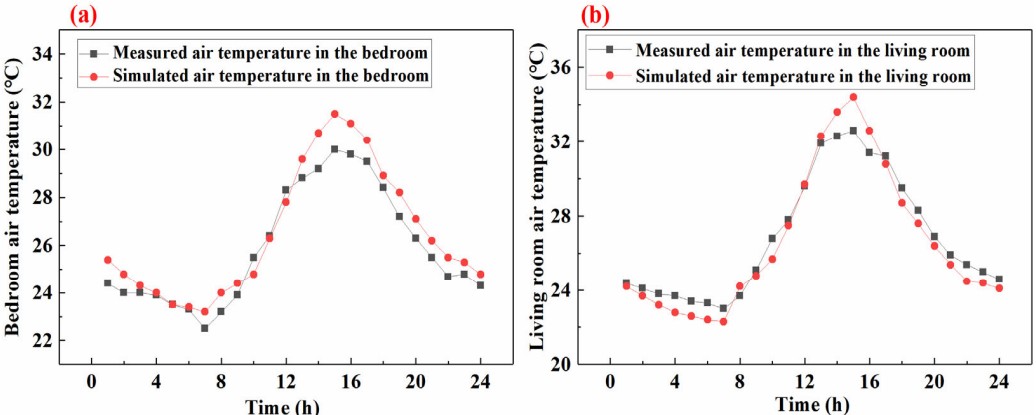

**Figure 7.** (**a**) Comparison between the measured data and the simulated data in the bedroom. (**b**) Comparison between the measured data and the simulated data in the living room.

Figure 7b shows the measured and simulated data for the living room in summer. The measured and simulated numerical curve trends of the living room were generally similar: the highest indoor air temperature appeared at 15:00 and the lowest at 7:00. The simulated average indoor air temperature of the living room is 26.5 °C, and the measured average indoor air temperature of the bedroom is 26.8 °C. The relative error between the simulated and measured values was only 1.1%. In summary, the simulation of the air temperature in the bedrooms and living rooms of traditional residential buildings was consistent with the measured results. Under the same outdoor climate conditions, the relative error between the simulation results and the measured values was less than 3%, which proves that the traditional house model (three-section courtyard) is highly reliable.

### 3.2. Comparison of Refrigeration Energy Consumption in the Renovation Scheme of the Main Material of the Wall

DesignBuilder was used to simulate the refrigeration energy consumption of the wall, and the simulation was conducted from 15 June to 1 August. The hottest week for refrigeration energy consumption was selected for comparison (26 July to 1 August). Figure 8 shows the building refrigeration energy consumption curves for Case 1 (solid clay bricks), Case 2 (porous clay bricks), and Case 3 (aerated concrete blocks). Among them, Case 1 had the highest refrigeration energy consumption, with a total energy consumption of 452.58 kW·h; Case 2 had a refrigeration energy consumption of 434 kW·h; and Case 3 had the lowest, with a total energy consumption of only 427.7 kW·h, which was 24.88 kW·h less than that of Case 1. This indicates that when the main material of the wall was an aerated concrete block, the wall-renovation scheme had the best thermal insulation effect. When the main material of the wall was solid clay brick, the thermal insulation effect of the wall renovation scheme was the poorest. The reason for this phenomenon is that the thermal resistance of aerated concrete blocks is higher than that of solid clay bricks; therefore, aerated concrete blocks as the main wall material (Case 3) can provide better insulation and reduce refrigeration energy consumption.

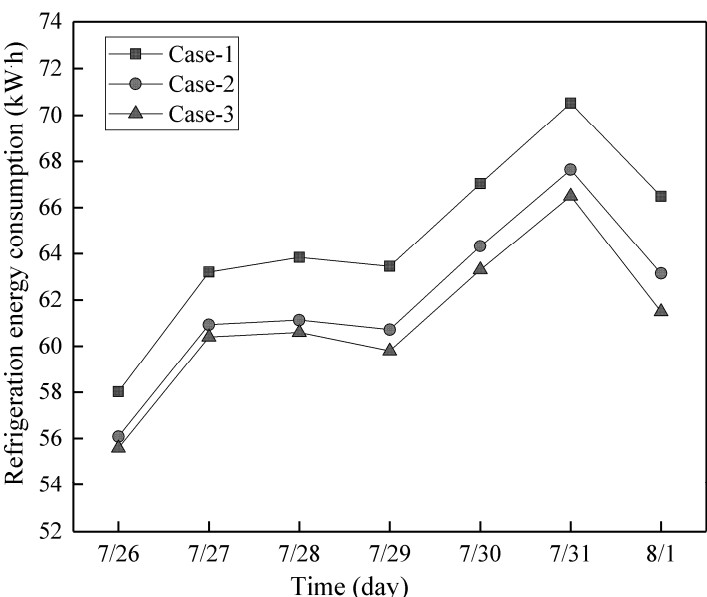

**Figure 8.** Comparison of the building refrigeration energy consumption of wall schemes with different primary materials.

### 3.3. Comparison of Refrigeration Energy Consumption of Wall Renovation Schemes with Added Air Interlayer

Figure 9 shows the building refrigeration energy consumption for Case 3 and Case 4 (added air interlayer). Between them, Case 3 had the highest refrigeration energy consumption, with a total energy consumption of 427.7 kW·h. The refrigeration energy consumption of Case 4 was 4.3 kW·h less than that of Case 3. This indicates that the wall-reconstruction scheme with an air interlayer (Case 4) had a better energy-saving effect. The reason for this phenomenon is that the air interlayer increases the thermal resistance of the wall, thus enhancing the thermal insulation ability of the wall. Therefore, Case 4 provided better insulation and consumed less refrigeration energy. In summary, the wall renovation scheme with the air interlayer can not only reduce energy consumption, but also enhance the thermal insulation ability of the wall, thus improving the indoor thermal environment, which also verifies the correctness of previous research results [20,21].

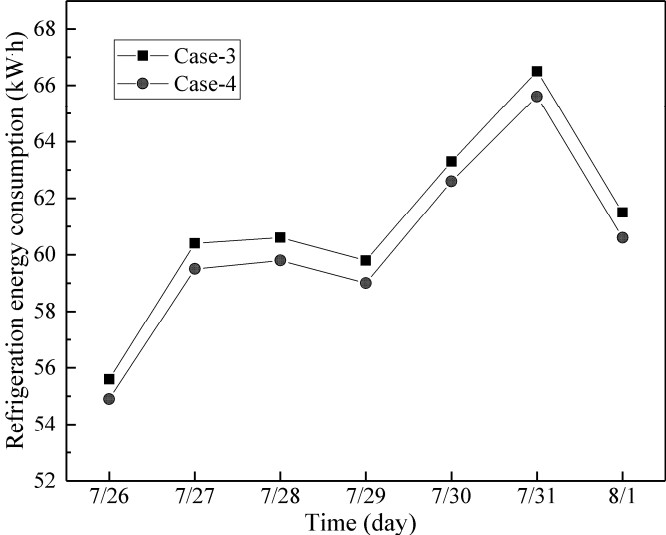

**Figure 9.** Comparison of building refrigeration energy consumption between Case 3 and Case 4.

### 3.4. Comparison of the Refrigeration Energy Consumption of Wall Renovation Schemes with Insulation Materials

Figure 10 compares the refrigeration energy consumption for Case 5 (mineral wool board), Case 6 (EPS), and Case 7 (XPS). After three kinds of sandwich materials of the same thickness are added to the external walls of the same material and thickness, the building refrigeration energy consumptions from highest to lowest were Case 5 (418 kW·h), Case 6 (413.1 kW·h), and Case 7 (409.8 kW·h). The above results show that the wall with insulation material had the function of reducing energy consumption and improving heat insulation. It also shows that the greater the thermal resistance of insulation materials, the better the thermal insulation effect, which also proves the correctness of the conclusions of Gou [18] and Hou [19] et al. In addition, Case 5 had the highest cooling energy consumption, which was 2.27% lower than that of Case 3 (without insulation materials). This indicates that Case 5's insulation effect and energy savings are poor. Case 7 had the lowest cooling energy consumption, which was 4.19% lower than that of Case 3 (without insulation). This phenomenon shows that Case 7 had the best thermal insulation effect and energy savings, which can provide a reference for traditional house designers.

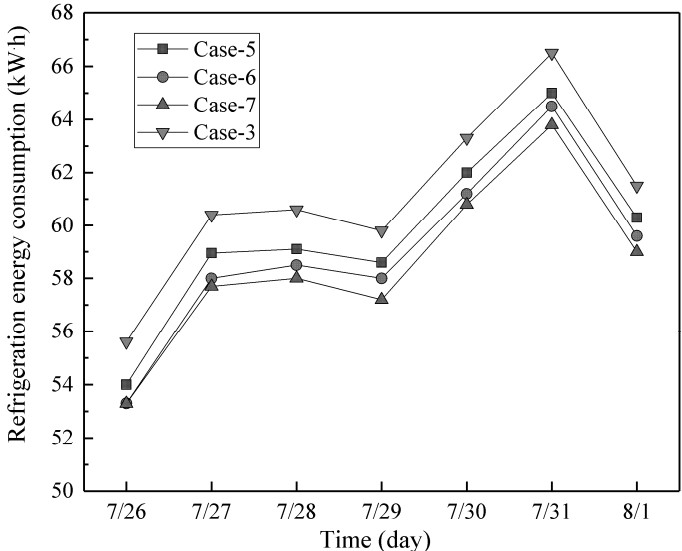

**Figure 10.** Comparison of the building refrigeration energy consumption of different insulation material schemes.

### 3.5. Comparative Analysis of the Refrigeration Energy Consumption of External Insulation and Internal Insulation Wall Renovation Schemes

Figure 11 shows the refrigeration energy consumption curves for the externally and internally insulated wall renovation schemes. The figure clearly shows that, after adding insulation materials to the external walls, the energy consumption of the building was significantly reduced. The reason is the same as the previous research conclusion [18–20], that is, the thermal resistance of the thermal insulation material is high, thus improving the thermal insulation effect of the external wall. In addition, Case 7 had the lowest cooling energy consumption, which was 4.19% lower than that of Case 3 (without insulation). Case 8's cooling energy consumption was 414.6 kW·h, which was 3.06% lower than that of Case 3 (without insulation). This phenomenon indicates that the thermal insulation effect and energy savings of Case 7 and Case 8 were better than those of Case 3, and the thermal insulation effect and energy-saving effect of Case 7 were better than those of Case 8. Therefore, by comparing Case 3, Case 7 and Case 8, it can be concluded that the thermal insulation effect of the middle part of the wall was better than that of the inner part, which also indicates that the thermal insulation scheme in the middle part of the wall has better thermal insulation and energy-saving effects in practical applications.

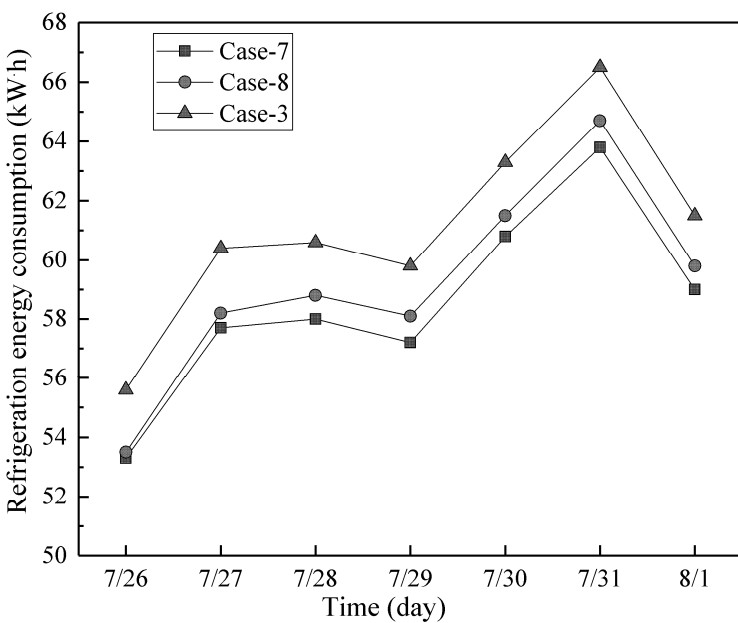

**Figure 11.** Comparison curve of the cooling energy consumption of different wall insulation methods.

## 4. Conclusions

To address the poor thermal environment of traditional residential buildings, which cannot meet the needs of building users for human thermal comfort, this study designed four sets of wall renovation plans based on the original traditional residential buildings and modern technology, and DesignBuilder was used to simulate and verify the feasibility of the building renovation schemes. The conclusions are as follows:

1.  Compared to Case 1 and Case 2, Case 3 had the lowest building cooling energy consumption, with a total refrigeration energy consumption of only 427.7 kW·h. This indicates that when the main material of the wall is an aerated concrete block, the wall-renovation scheme has the best heat insulation and energy-saving effects.

2.  The refrigeration energy consumption of Case 4 in the hottest week was 422 kW·h, which was 4.3 kW·h less than that of Case 3. This indicates that the wall-renovation scheme with air interlayers exhibits better thermal insulation and energy-saving effects. This study serves as a reference for traditional house designers.

3.  Compared with Case 5 and Case 6, Case 7 consumed the least amount of energy in the hottest week at only 409.8 kW·h. Therefore, the XPS thermal insulation material transformation scheme has better thermal insulation and energy-saving effects.

4.  The refrigeration energy consumption of Case 7 was only 409.8 kW·h in the hottest week, which was 4.19% lower than that of Case 3 (without insulation materials). Therefore, selecting a central insulation scheme for an actual project results in better thermal insulation and energy-saving effects.

In summary, the use of aerated concrete blocks, the addition of an air interlayer, XPS insulation materials, and a central insulation scheme can improve the thermal environment and human thermal comfort in traditional residential buildings. However, this study did not determine whether the above renovation scheme can be adapted to traditional residential buildings in winter climate conditions. Therefore, future research should focus on the applicability of the indoor thermal environment of traditional residential buildings under winter climatic conditions.

**Author Contributions:** Conceptualization, W.G.; Methodology, C.H., W.H. and W.G.; Software, Y.J.; Formal analysis, C.H. and W.H.; Resources, W.H.; Writing—original draft, C.H. and Y.J.; Writing—review & editing, C.H. and W.H.; Visualization, Y.J. All authors have read and agreed to the published version of the manuscript.

**Funding:** This research received no external funding.

**Institutional Review Board Statement:** Not applicable.

**Informed Consent Statement:** Not applicable.

**Data Availability Statement:** Data are contained within the article.

**Conflicts of Interest:** The authors declare no conflicts of interest.

## Nomenclatures

| | |
|---|---|
| $q''_{\alpha sol}$ | Absorption of direct and diffuse solar (short wavelength) radiation by heat flow, [W/m$^2$]; |
| $q''_{LWR}$ | The exchange of heat radiation flux with air and the surrounding environment, [W/m$^2$]; |
| $q''_{outdoor}$ | Convection flux exchange between the outer surface and the outside air, [W/m$^2$]; |
| $q''_{ko}$ | Heat conduction flux (q/A) to the wall surface, [W/m$^2$]; |
| $q_i$ | The heat exchange intensity per unit time and area of the inner surface of the wall, [W/m$^2$]; |
| $q_{ic}$ | The heat transferred by air to the inner surface of the wall by convective heat transfer in unit time, [W/m$^2$]. |
| $q_{ir}$ | The heat transferred to the inner surface of the wall by means of radiant heat exchange on each surface of the heat tank in unit time, [W/m$^2$]. |
| $\alpha_i$ | Heat transfer coefficient of the inner surface of the wall, [W/(m$^2$·K)]. |
| $T_{in}$ | Air temperature in the hot box, [°C]. |
| $T_j$ | The inner surface temperature of the wall, [°C]. |
| $T_i$ | The outer surface temperature of the wall, [°C]. |
| $T_{out}$ | Air temperature in a cold box, [°C]. |
| $q_y$ | Intensity of heat flow through the wall per unit time and per unit area, [W/m$^2$]. |
| $R$ | Thermal conductivity resistance of the wall, [m$^2$·K/W]. |
| $q_e$ | Heat flow intensity on the surface of the wall per unit area per unit time, [W/m$^2$]. |
| $\alpha_e$ | Heat transfer coefficient of the outer surface of the wall, [W/(m$^2$·K)]. |
| $q''_{LWX}$ | Net long-wave radiation energy exchanged with the room surface, [W/m$^2$]; |
| $q''_{SW}$ | Net shortwave radiation energy from light emission, [W/m$^2$]; |
| $q''_{LWS}$ | The amount of long-wave radiation emitted by equipment in the room, [W/m$^2$]; |
| $q''_{ki}$ | Heat gain through the wall, [W/m$^2$]; |
| $q''_{sol}$ | Solar radiation heat absorbed by the wall surface, [W/m$^2$]; |
| $q''_{indoor}$ | Convection heat transfer with indoor air, [W/m$^2$]. |

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
