# Peer review of "Optimization and Renovation Design of Indoor Thermal Environment in Traditional Houses in Northeast Sichuan (China)—A Case Study of a Three-Section Courtyard House"

_sustainability, doi:10.3390/su16072921_

Round 1

Reviewer 1 Report

Comments and Suggestions for Authors

In order to meet the building's demand for human thermal comfort and optimize the indoor thermal environment of traditional residential buildings, Hou and his colleagues designed 4 groups of wall renovation schemes based on the original buildings of traditional residential buildings and modern technologies, and simulated and verified the feasibility of 18 building renovation schemes by using Design Builder. It is worth noting that although the results show that the wall renovation scheme using air sandwich has better thermal insulation and energy saving effects, it does not fundamentally and technically solve the problem of building users' needs for human thermal comfort, and the building structure has not been improved. In general, I think this work is not suitable for this journal, and suggest that it be voted for more specialized journals.

Additional comments:

1, Figure 1, Figure 7 of the graphics clarity is low, can improve the resolution of the graphics?

2. In the Results and discussion of the second part and the third part, there are no references and scientific analysis.

3. In the first part of Introduction, although there are some references, the references are relatively old, there is no latest cutting-edge technology reference, and there is a lack of innovative research.

4. Page 7, Formula (1), formula (2), formula (3), formula (4) and formula (5) on page 8 can be organized and written by the formula editor.

Author Response

All reviewer′ comments are truly appreciated. Please check the PDF versions for the reply manuscript.

Reviewer 2 Report

Comments and Suggestions for Authors

First of all I would like to congratulate the authors, it is certainly an article of interest for the field of optimization and renovations design considering thermal envelopes. Although the article has an adequate development, I observe some points in which I recommend the authors to deepen a little more, to get a final document of contribution to the research.

Abstract:

Although in general lines the abstract allows us to have a first approximation of the research, I recommend to the authors that 1) a clearer and more specific explanation about the methodology used and 2) I do not know the implications of the research.

Introduction:

The research gap is not clearly defined, it would be of interest to go a little deeper into the subject.

The bibliography is not updated, only two recent articles of the total bibliography are identified, and considering the important advances in recent years in the field of sustainable construction and the environment, it would be of interest to update the bibliography with recent publications that other authors have made on the subject of the article.

The working structure that the article will follow is not anticipated.

Discussion and results:

I do not see a direct relationship between the results obtained and the bibliography consulted. This point is always of interest, to be able to find the meeting point between both sessions of an article in order to analyze the results.

Conclusions:

It is not clear what the conclusions are based on the paper's contribution.

Similarly, the limitations of the research and future directions are not elaborated upon.

Finally, I would like to encourage the authors to take into account the proposed revisions in order to improve the quality of the article.

Author Response

(The authors gave the same response as above.)

Reviewer 3 Report

Comments and Suggestions for Authors

The reviewer believes that the work presented by the authors is of great interest to the scientific community. The structure of the article is correct and contributes to provide a good thread to its reading.

The reviewer considers that the authors perfectly present the motivation and the problematic that leads to the research with a very well directed and presented introduction; emphasising and mentioning the current situation, the background and the already existing contributions on the subject under study. The methodology and results are clearly presented; this makes it easier for the reader to understand the research. Finally, the conclusions of all case combinations are presented with quantitative information that clearly reinforces the rationale for the conclusions. It is possible that these conclusions could be reinforced by comparing the results of the proposed methodology with others cited by the authors, but it is also true that this itself could constitute new research that would encourage the authors to continue this work.    

Author Response

(The authors gave the same response as above.)

Round 2

Reviewer 1 Report

Comments and Suggestions for Authors

In order to meet the building's demand for human thermal comfort and optimize the indoor thermal environment of traditional residential buildings, Hou and his colleagues designed 4 groups of wall renovation schemes based on the original buildings of traditional residential buildings and modern technologies, and simulated and verified the feasibility of 18 building renovation schemes by using Design Builder. It is worth noting that although the results show that the wall renovation scheme using air sandwich has better thermal insulation and energy saving effects, it does not fundamentally and technically solve the problem of building users' needs for human thermal comfort, and the building structure has not been improved. In general, I think this work is not suitable for this journal, and suggest that it be voted for more specialized journals.

Additional comments:

1, Figure 1, Figure 7 of the graphics clarity is low, can improve the resolution of the graphics?

2. In the Results and discussion of the second part and the third part, there are no references and scientific analysis.

3. In the first part of Introduction, although there are some references, the references are relatively old, there is no latest cutting-edge technology reference, and there is a lack of innovative research.

4. Page 7 ,Formula (1), formula (2), formula (3), formula (4) and formula (5) on page 8 can be organized and written by the formula editor.

Comments on the Quality of English Language

In order to meet the building's demand for human thermal comfort and optimize the indoor thermal environment of traditional residential buildings, Hou and his colleagues designed 4 groups of wall renovation schemes based on the original buildings of traditional residential buildings and modern technologies, and simulated and verified the feasibility of 18 building renovation schemes by using Design Builder. It is worth noting that although the results show that the wall renovation scheme using air sandwich has better thermal insulation and energy saving effects, it does not fundamentally and technically solve the problem of building users' needs for human thermal comfort, and the building structure has not been improved. In general, I think this work is not suitable for this journal, and suggest that it be voted for more specialized journals.

Additional comments:

1, Figure 1, Figure 7 of the graphics clarity is low, can improve the resolution of the graphics?

2. In the Results and discussion of the second part and the third part, there are no references and scientific analysis.

3. In the first part of Introduction, although there are some references, the references are relatively old, there is no latest cutting-edge technology reference, and there is a lack of innovative research.

4. Page 7 ,Formula (1), formula (2), formula (3), formula (4) and formula (5) on page 8 can be organized and written by the formula editor.

Author Response

Firstly, thanks for your positive evaluation and all your comments, which were extremely valuable in helping to improve the paper quality. According to your comments, we tried our best to make the appropriate revision. In addition, We have revised and checked the English expression and typos of this manuscript. English writing is polished by a native speaker, and obtain the certificate of polishing. (Please check the PDF versions for the reply manuscript.)

Reviewer 2 Report

Comments and Suggestions for Authors

I would like to congratulate the authors for taking into account the comments previously made to improve the document. 

Author Response

Thank you for your recognition of my work. Thanks for your positive evaluation and all your comments.

Round 3

Reviewer 1 Report

Comments and Suggestions for Authors

I have reviewed this paper before, and I could confirm that the author has considered some of the previous comments and made relatively good revisions. However, I think the author should improve the clarity of the picture. In conclusion, it seems to me that this study might be accepted by Sustainability.